# Improved Training of Wasserstein GANs

**Ishaan Gulrajani**[1]*,**Faruk Ahmed**[1], **Martin Arjovsky**[2], **Vincent Dumoulin**[1], **Aaron Courville**[1,3]
[1] Montreal Institute for Learning Algorithms
[2] Courant Institute of Mathematical Sciences
[3] CIFAR Fellow
igul222@gmail.com
{faruk.ahmed,vincent.dumoulin,aaron.courville}@umontreal.ca
ma4371@nyu.edu

## Abstract

Generative Adversarial Networks (GANs) are powerful generative models, but suffer from training instability. The recently proposed Wasserstein GAN (WGAN) makes progress toward stable training of GANs, but sometimes can still generate only poor samples or fail to converge. We find that these problems are often due to the use of weight clipping in WGAN to enforce a Lipschitz constraint on the critic, which can lead to undesired behavior. We propose an alternative to clipping weights: penalize the norm of gradient of the critic with respect to its input. Our proposed method performs better than standard WGAN and enables stable training of a wide variety of GAN architectures with almost no hyperparameter tuning, including 101-layer ResNets and language models with continuous generators. We also achieve high quality generations on CIFAR-10 and LSUN bedrooms. [†]

## 1 Introduction

Generative Adversarial Networks (GANs) [9] are a powerful class of generative models that cast generative modeling as a game between two networks: a generator network produces synthetic data given some noise source and a discriminator network discriminates between the generator's output and true data. GANs can produce very visually appealing samples, but are often hard to train, and much of the recent work on the subject [22, 18, 2, 20] has been devoted to finding ways of stabilizing training. Despite this, consistently stable training of GANs remains an open problem.

In particular, [1] provides an analysis of the convergence properties of the value function being optimized by GANs. Their proposed alternative, named Wasserstein GAN (WGAN) [2], leverages the Wasserstein distance to produce a value function which has better theoretical properties than the original. WGAN requires that the discriminator (called the *critic* in that work) must lie within the space of 1-Lipschitz functions, which the authors enforce through weight clipping.

Our contributions are as follows:

1. On toy datasets, we demonstrate how critic weight clipping can lead to undesired behavior.

2. We propose *gradient penalty (WGAN-GP)*, which does not suffer from the same problems.

3. We demonstrate stable training of varied GAN architectures, performance improvements over weight clipping, high-quality image generation, and a character-level GAN language model without any discrete sampling.

[†]Code for our models is available at https://github.com/igul222/improved_wgan_training.

## 2 Background

### 2.1 Generative adversarial networks

The GAN training strategy is to define a game between two competing networks. The *generator* network maps a source of noise to the input space. The *discriminator* network receives either a generated sample or a true data sample and must distinguish between the two. The generator is trained to fool the discriminator.

Formally, the game between the generator $G$ and the discriminator $D$ is the minimax objective:

$$\min_G \max_D \mathbb{E}_{\boldsymbol{x} \sim \mathbb{P}_r}[\log(D(\boldsymbol{x}))] + \mathbb{E}_{\tilde{\boldsymbol{x}} \sim \mathbb{P}_g}[\log(1 - D(\tilde{\boldsymbol{x}}))], \tag{1}$$

where $\mathbb{P}_r$ is the data distribution and $\mathbb{P}_g$ is the model distribution implicitly defined by $\tilde{\boldsymbol{x}} = G(\boldsymbol{z})$, $\boldsymbol{z} \sim p(\boldsymbol{z})$ (the input $\boldsymbol{z}$ to the generator is sampled from some simple noise distribution, such as the uniform distribution or a spherical Gaussian distribution).

If the discriminator is trained to optimality before each generator parameter update, then minimizing the value function amounts to minimizing the Jensen-Shannon divergence between $\mathbb{P}_r$ and $\mathbb{P}_g$ [9], but doing so often leads to vanishing gradients as the discriminator saturates. In practice, [9] advocates that the generator be instead trained to maximize $\mathbb{E}_{\tilde{\boldsymbol{x}} \sim \mathbb{P}_g}[\log(D(\tilde{\boldsymbol{x}}))]$, which goes some way to circumvent this difficulty. However, even this modified loss function can misbehave in the presence of a good discriminator [1].

### 2.2 Wasserstein GANs

[2] argues that the divergences which GANs typically minimize are potentially not continuous with respect to the generator's parameters, leading to training difficulty. They propose instead using the *Earth-Mover* (also called Wasserstein-1) distance $W(q, p)$, which is informally defined as the minimum cost of transporting mass in order to transform the distribution $q$ into the distribution $p$ (where the cost is mass times transport distance). Under mild assumptions, $W(q, p)$ is continuous everywhere and differentiable almost everywhere.

The WGAN value function is constructed using the Kantorovich-Rubinstein duality [24] to obtain

$$\min_G \max_{D \in \mathcal{D}} \mathbb{E}_{\boldsymbol{x} \sim \mathbb{P}_r}\left[D(\boldsymbol{x})\right] - \mathbb{E}_{\tilde{\boldsymbol{x}} \sim \mathbb{P}_g}\left[D(\tilde{\boldsymbol{x}}))\right] \tag{2}$$

where $\mathcal{D}$ is the set of 1-Lipschitz functions and $\mathbb{P}_g$ is once again the model distribution implicitly defined by $\tilde{\boldsymbol{x}} = G(\boldsymbol{z})$, $\boldsymbol{z} \sim p(\boldsymbol{z})$. In that case, under an optimal discriminator (called a *critic* in the paper, since it's not trained to classify), minimizing the value function with respect to the generator parameters minimizes $W(\mathbb{P}_r, \mathbb{P}_g)$.

The WGAN value function results in a critic function whose gradient with respect to its input is better behaved than its GAN counterpart, making optimization of the generator easier. Additionally, WGAN has the desirable property that its value function correlates with sample quality, which is not the case for GANs.

To enforce the Lipschitz constraint on the critic, [2] propose to clip the weights of the critic to lie within a compact space $[-c, c]$. The set of functions satisfying this constraint is a subset of the $k$-Lipschitz functions for some $k$ which depends on $c$ and the critic architecture. In the following sections, we demonstrate some of the issues with this approach and propose an alternative.

### 2.3 Properties of the optimal WGAN critic

In order to understand why weight clipping is problematic in a WGAN critic, as well as to motivate our approach, we highlight some properties of the optimal critic in the WGAN framework. We prove these in the Appendix.

**Proposition 1.** *Let $\mathbb{P}_r$ and $\mathbb{P}_g$ be two distributions in $\mathcal{X}$, a compact metric space. Then, there is a 1-Lipschitz function $f^*$ which is the optimal solution of $\max_{\|f\|_L \leq 1} \mathbb{E}_{y \sim \mathbb{P}_r}[f(y)] - \mathbb{E}_{x \sim \mathbb{P}_g}[f(x)]$. Let $\pi$ be the optimal coupling between $\mathbb{P}_r$ and $\mathbb{P}_g$, defined as the minimizer of: $W(\mathbb{P}_r, \mathbb{P}_g) = \inf_{\pi \in \Pi(\mathbb{P}_r, \mathbb{P}_g)} \mathbb{E}_{(x,y) \sim \pi}[\|x - y\|]$ where $\Pi(\mathbb{P}_r, \mathbb{P}_g)$ is the set of joint distributions $\pi(x, y)$ whose marginals are $\mathbb{P}_r$ and $\mathbb{P}_g$, respectively. Then, if $f^*$ is differentiable[‡], $\pi(x = y) = 0$[§], and $x_t = tx + (1-t)y$ with $0 \leq t \leq 1$, it holds that $\mathbb{P}_{(x,y) \sim \pi}\left[\nabla f^*(x_t) = \frac{y - x_t}{\|y - x_t\|}\right] = 1$.*

**Corollary 1.** *$f^*$ has gradient norm 1 almost everywhere under $\mathbb{P}_r$ and $\mathbb{P}_g$.*

## 3 Difficulties with weight constraints

We find that weight clipping in WGAN leads to optimization difficulties, and that even when optimization succeeds the resulting critic can have a pathological value surface. We explain these problems below and demonstrate their effects; however we do not claim that each one always occurs in practice, nor that they are the only such mechanisms.

Our experiments use the specific form of weight constraint from [2] (hard clipping of the magnitude of each weight), but we also tried other weight constraints (L2 norm clipping, weight normalization), as well as soft constraints (L1 and L2 weight decay) and found that they exhibit similar problems.

To some extent these problems can be mitigated with batch normalization in the critic, which [2] use in all of their experiments. However even with batch normalization, we observe that very deep WGAN critics often fail to converge.

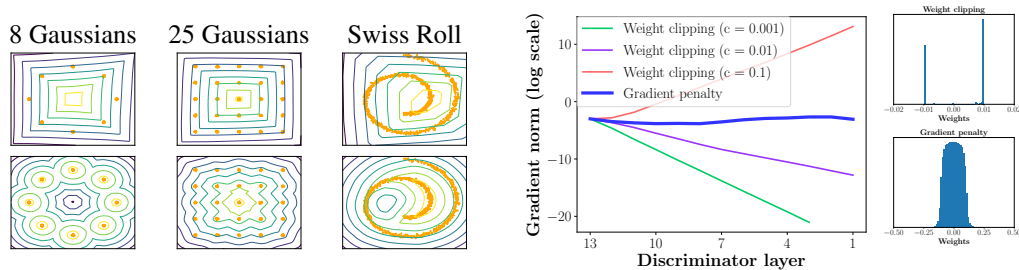

(a) Value surfaces of WGAN critics trained to optimality on toy datasets using (top) weight clipping and (bottom) gradient penalty. Critics trained with weight clipping fail to capture higher moments of the data distribution. The 'generator' is held fixed at the real data plus Gaussian noise.

(b) (left) Gradient norms of deep WGAN critics during training on toy datasets either explode or vanish when using weight clipping, but not when using a gradient penalty. (right) Weight clipping (top) pushes weights towards two values (the extremes of the clipping range), unlike gradient penalty (bottom).

Figure 1: Gradient penalty in WGANs does not exhibit undesired behavior like weight clipping.

### 3.1 Capacity underuse

Implementing a $k$-Lipshitz constraint via weight clipping biases the critic towards much simpler functions. As stated previously in Corollary 1, the optimal WGAN critic has unit gradient norm almost everywhere under $\mathbb{P}_r$ and $\mathbb{P}_g$; under a weight-clipping constraint, we observe that our neural network architectures which try to attain their maximum gradient norm $k$ end up learning extremely simple functions.

To demonstrate this, we train WGAN critics with weight clipping to optimality on several toy distributions, holding the generator distribution $\mathbb{P}_g$ fixed at the real distribution plus unit-variance Gaussian noise. We plot value surfaces of the critics in Figure 1a. We omit batch normalization in the

---

[‡]We can actually assume much less, and talk only about directional derivatives on the direction of the line; which we show in the proof always exist. This would imply that in every point where $f^*$ is differentiable (and thus we can take gradients in a neural network setting) the statement holds.

[§]This assumption is in order to exclude the case when the matching point of sample $x$ is $x$ itself. It is satisfied in the case that $\mathbb{P}_r$ and $\mathbb{P}_g$ have supports that intersect in a set of measure 0, such as when they are supported by two low dimensional manifolds that don't perfectly align [1].

**Algorithm 1** WGAN with gradient penalty. We use default values of $\lambda = 10$, $n_{\text{critic}} = 5$, $\alpha = 0.0001$, $\beta_1 = 0$, $\beta_2 = 0.9$.

---

**Require:** The gradient penalty coefficient $\lambda$, the number of critic iterations per generator iteration $n_{\text{critic}}$, the batch size $m$, Adam hyperparameters $\alpha, \beta_1, \beta_2$.
**Require:** initial critic parameters $w_0$, initial generator parameters $\theta_0$.

1: **while** $\theta$ has not converged **do**
2:     **for** $t = 1, ..., n_{\text{critic}}$ **do**
3:         **for** $i = 1, ..., m$ **do**
4:             Sample real data $\boldsymbol{x} \sim \mathbb{P}_r$, latent variable $\boldsymbol{z} \sim p(\boldsymbol{z})$, a random number $\epsilon \sim U[0, 1]$.
5:             $\tilde{\boldsymbol{x}} \leftarrow G_\theta(\boldsymbol{z})$
6:             $\hat{\boldsymbol{x}} \leftarrow \epsilon \boldsymbol{x} + (1 - \epsilon)\tilde{\boldsymbol{x}}$
7:             $L^{(i)} \leftarrow D_w(\tilde{\boldsymbol{x}}) - D_w(\boldsymbol{x}) + \lambda(\|\nabla_{\hat{\boldsymbol{x}}} D_w(\hat{\boldsymbol{x}})\|_2 - 1)^2$
8:         **end for**
9:         $w \leftarrow \text{Adam}(\nabla_w \frac{1}{m} \sum_{i=1}^{m} L^{(i)}, w, \alpha, \beta_1, \beta_2)$
10:     **end for**
11:     Sample a batch of latent variables $\{\boldsymbol{z}^{(i)}\}_{i=1}^{m} \sim p(\boldsymbol{z})$.
12:     $\theta \leftarrow \text{Adam}(\nabla_\theta \frac{1}{m} \sum_{i=1}^{m} -D_w(G_\theta(\boldsymbol{z})), \theta, \alpha, \beta_1, \beta_2)$
13: **end while**

---

critic. In each case, the critic trained with weight clipping ignores higher moments of the data distribution and instead models very simple approximations to the optimal functions. In contrast, our approach does not suffer from this behavior.

### 3.2 Exploding and vanishing gradients

We observe that the WGAN optimization process is difficult because of interactions between the weight constraint and the cost function, which result in either vanishing or exploding gradients without careful tuning of the clipping threshold $c$.

To demonstrate this, we train WGAN on the Swiss Roll toy dataset, varying the clipping threshold $c$ in $[10^{-1}, 10^{-2}, 10^{-3}]$, and plot the norm of the gradient of the critic loss with respect to successive layers of activations. Both generator and critic are 12-layer ReLU MLPs without batch normalization. Figure 1b shows that for each of these values, the gradient either grows or decays exponentially as we move farther back in the network. We find our method results in more stable gradients that neither vanish nor explode, allowing training of more complicated networks.

## 4 Gradient penalty

We now propose an alternative way to enforce the Lipschitz constraint. A differentiable function is 1-Lipschtiz if and only if it has gradients with norm at most 1 everywhere, so we consider directly constraining the gradient norm of the critic's output with respect to its input. To circumvent tractability issues, we enforce a soft version of the constraint with a penalty on the gradient norm for random samples $\hat{\boldsymbol{x}} \sim \mathbb{P}_{\hat{\boldsymbol{x}}}$. Our new objective is

$$L = \underbrace{\mathbb{E}_{\tilde{\boldsymbol{x}} \sim \mathbb{P}_g}[D(\tilde{\boldsymbol{x}})] - \mathbb{E}_{\boldsymbol{x} \sim \mathbb{P}_r}[D(\boldsymbol{x})]}_{\text{Original critic loss}} + \underbrace{\lambda \mathbb{E}_{\hat{\boldsymbol{x}} \sim \mathbb{P}_{\hat{\boldsymbol{x}}}}\left[(\|\nabla_{\hat{\boldsymbol{x}}} D(\hat{\boldsymbol{x}})\|_2 - 1)^2\right]}_{\text{Our gradient penalty}}. \tag{3}$$

**Sampling distribution** We implicitly define $\mathbb{P}_{\hat{\boldsymbol{x}}}$ sampling uniformly along straight lines between pairs of points sampled from the data distribution $\mathbb{P}_r$ and the generator distribution $\mathbb{P}_g$. This is motivated by the fact that the optimal critic contains straight lines with gradient norm 1 connecting coupled points from $\mathbb{P}_r$ and $\mathbb{P}_g$ (see Proposition 1). Given that enforcing the unit gradient norm constraint everywhere is intractable, enforcing it only along these straight lines seems sufficient and experimentally results in good performance.

**Penalty coefficient** All experiments in this paper use $\lambda = 10$, which we found to work well across a variety of architectures and datasets ranging from toy tasks to large ImageNet CNNs.

**No critic batch normalization** Most prior GAN implementations [21, 22, 2] use batch normalization in both the generator and the discriminator to help stabilize training, but batch normalization changes the form of the discriminator's problem from mapping a single input to a single output to mapping from an entire batch of inputs to a batch of outputs [22]. Our penalized training objective is no longer valid in this setting, since we penalize the norm of the critic's gradient with respect to each input independently, and not the entire batch. To resolve this, we simply omit batch normalization in the critic in our models, finding that they perform well without it. Our method works with normalization schemes which don't introduce correlations between examples. In particular, we recommend layer normalization [3] as a drop-in replacement for batch normalization.

**Two-sided penalty** We encourage the norm of the gradient to go towards 1 (two-sided penalty) instead of just staying below 1 (one-sided penalty). Empirically this seems not to constrain the critic too much, likely because the optimal WGAN critic anyway has gradients with norm 1 almost everywhere under $\mathbb{P}_r$ and $\mathbb{P}_g$ and in large portions of the region in between (see subsection 2.3). In our early observations we found this to perform slightly better, but we don't investigate this fully. We describe experiments on the one-sided penalty in the appendix.

## 5 Experiments

### 5.1 Training random architectures within a set

We experimentally demonstrate our model's ability to train a large number of architectures which we think are useful to be able to train. Starting from the DCGAN architecture, we define a set of architecture variants by changing model settings to random corresponding values in Table 1. We believe that reliable training of many of the architectures in this set is a useful goal, but we do not claim that our set is an unbiased or representative sample of the whole space of useful architectures: it is designed to demonstrate a successful regime of our method, and readers should evaluate whether it contains architectures similar to their intended application.

Table 1: We evaluate WGAN-GP's ability to train the architectures in this set.

| | |
|---|---|
| Nonlinearity ($G$) | [ReLU, LeakyReLU, $\frac{\text{softplus}(2x+2)}{2} - 1, tanh$] |
| Nonlinearity ($D$) | [ReLU, LeakyReLU, $\frac{\text{softplus}(2x+2)}{2} - 1, tanh$] |
| Depth ($G$) | [4, 8, 12, 20] |
| Depth ($D$) | [4, 8, 12, 20] |
| Batch norm ($G$) | [True, False] |
| Batch norm ($D$; layer norm for WGAN-GP) | [True, False] |
| Base filter count ($G$) | [32, 64, 128] |
| Base filter count ($D$) | [32, 64, 128] |

From this set, we sample 200 architectures and train each on $32 \times 32$ ImageNet with both WGAN-GP and the standard GAN objectives. Table 2 lists the number of instances where either: only the standard GAN succeeded, only WGAN-GP succeeded, both succeeded, or both failed, where success is defined as `inception_score > min_score`. For most choices of score threshold, WGAN-GP successfully trains many architectures from this set which we were unable to train with the standard GAN objective.

Table 2: Outcomes of training 200 random architectures, for different success thresholds. For comparison, our standard DCGAN achieved a score of 7.24. A longer version of this table can be found in the appendix.

| Min. score | Only GAN | Only WGAN-GP | Both succeeded | Both failed |
|---|---|---|---|---|
| 1.0 | 0 | 8 | 192 | 0 |
| 3.0 | 1 | 88 | 110 | 1 |
| 5.0 | 0 | 147 | 42 | 11 |
| 7.0 | 1 | 104 | 5 | 90 |
| 9.0 | 0 | 0 | 0 | 200 |

| DCGAN | LSGAN | WGAN (clipping) | WGAN-GP (ours) |
|---|---|---|---|

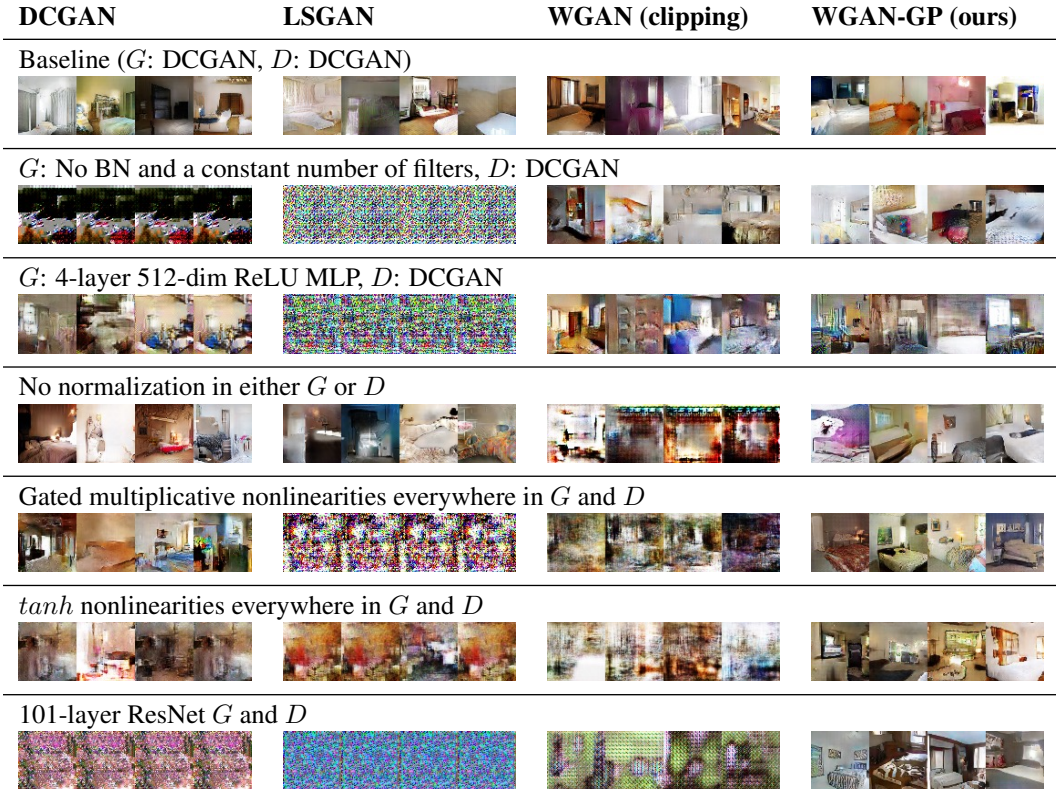

Figure 2: Different GAN architectures trained with different methods. We only succeeded in training every architecture with a shared set of hyperparameters using WGAN-GP.

## 5.2 Training varied architectures on LSUN bedrooms

To demonstrate our model's ability to train many architectures with its default settings, we train six different GAN architectures on the LSUN bedrooms dataset [30]. In addition to the baseline DC-GAN architecture from [21], we choose six architectures whose successful training we demonstrate: *(1)* no BN and a constant number of filters in the generator, as in [2], *(2)* 4-layer 512-dim ReLU MLP generator, as in [2], *(3)* no normalization in either the discriminator or generator *(4)* gated multiplicative nonlinearities, as in [23], *(5)* $tanh$ nonlinearities, and *(6)* 101-layer ResNet generator and discriminator.

Although we do not claim it is impossible without our method, to the best of our knowledge this is the first time very deep residual networks were successfully trained in a GAN setting. For each architecture, we train models using four different GAN methods: WGAN-GP, WGAN with weight clipping, DCGAN [21], and Least-Squares GAN [17]. For each objective, we used the default set of optimizer hyperparameters recommended in that work (except LSGAN, where we searched over learning rates).

For WGAN-GP, we replace any batch normalization in the discriminator with layer normalization (see section 4). We train each model for 200K iterations and present samples in Figure 2. We only succeeded in training every architecture with a shared set of hyperparameters using WGAN-GP. For every other training method, some of these architectures were unstable or suffered from mode collapse.

## 5.3 Improved performance over weight clipping

One advantage of our method over weight clipping is improved training speed and sample quality. To demonstrate this, we train WGANs with weight clipping and our gradient penalty on CIFAR-10 [13] and plot Inception scores [22] over the course of training in Figure 3. For WGAN-GP,

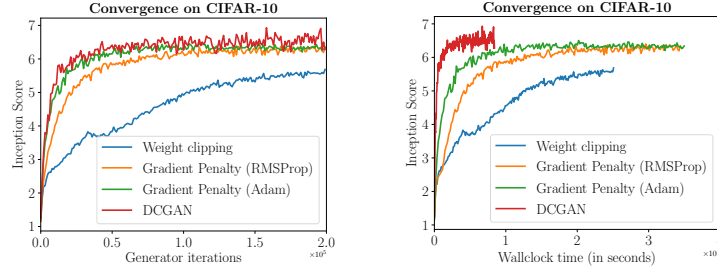

Figure 3: CIFAR-10 Inception score over generator iterations (left) or wall-clock time (right) for four models: WGAN with weight clipping, WGAN-GP with RMSProp and Adam (to control for the optimizer), and DCGAN. WGAN-GP significantly outperforms weight clipping and performs comparably to DCGAN.

we train one model with the same optimizer (RMSProp) and learning rate as WGAN with weight clipping, and another model with Adam and a higher learning rate. Even with the same optimizer, our method converges faster and to a better score than weight clipping. Using Adam further improves performance. We also plot the performance of DCGAN [21] and find that our method converges more slowly (in wall-clock time) than DCGAN, but its score is more stable at convergence.

## 5.4 Sample quality on CIFAR-10 and LSUN bedrooms

For equivalent architectures, our method achieves comparable sample quality to the standard GAN objective. However the increased stability allows us to improve sample quality by exploring a wider range of architectures. To demonstrate this, we find an architecture which establishes a new state of the art Inception score on unsupervised CIFAR-10 (Table 3). When we add label information (using the method in [19]), the same architecture outperforms all other published models except for SGAN.

Table 3: Inception scores on CIFAR-10. Our unsupervised model achieves state-of-the-art performance, and our conditional model outperforms all others except SGAN.

| Unsupervised | | Supervised | |
| --- | --- | --- | --- |
| Method | Score | Method | Score |
| ALI [8] (in [26]) | $5.34 \pm .05$ | SteinGAN [25] | 6.35 |
| BEGAN [4] | 5.62 | DCGAN (with labels, in [25]) | 6.58 |
| DCGAN [21] (in [11]) | $6.16 \pm .07$ | Improved GAN [22] | $8.09 \pm .07$ |
| Improved GAN (-L+HA) [22] | $6.86 \pm .06$ | AC-GAN [19] | $8.25 \pm .07$ |
| EGAN-Ent-VI [7] | $7.07 \pm .10$ | SGAN-no-joint [11] | $8.37 \pm .08$ |
| DFM [26] | $7.72 \pm .13$ | WGAN-GP ResNet (ours) | $8.42 \pm .10$ |
| **WGAN-GP ResNet (ours)** | $7.86 \pm .07$ | **SGAN** [11] | $8.59 \pm .12$ |

We also train a deep ResNet on $128 \times 128$ LSUN bedrooms and show samples in Figure 4. We believe these samples are at least competitive with the best reported so far on any resolution for this dataset.

## 5.5 Modeling discrete data with a continuous generator

To demonstrate our method's ability to model degenerate distributions, we consider the problem of modeling a complex discrete distribution with a GAN whose generator is defined over a continuous space. As an instance of this problem, we train a character-level GAN language model on the Google Billion Word dataset [6]. Our generator is a simple 1D CNN which deterministically transforms a latent vector into a sequence of 32 one-hot character vectors through 1D convolutions. We apply a softmax nonlinearity at the output, but use no sampling step: during training, the softmax output is

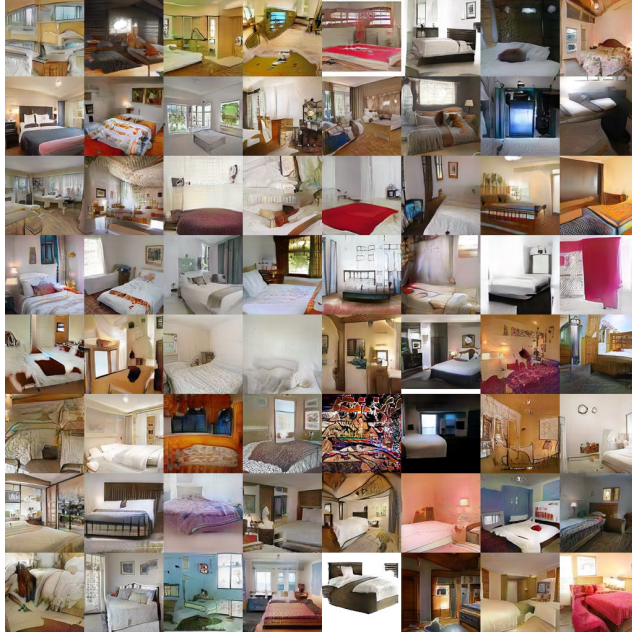

Figure 4: Samples of $128 \times 128$ LSUN bedrooms. We believe these samples are at least comparable to the best published results so far.

passed directly into the critic (which, likewise, is a simple 1D CNN). When decoding samples, we just take the argmax of each output vector.

We present samples from the model in Table 4. Our model makes frequent spelling errors (likely because it has to output each character independently) but nonetheless manages to learn quite a lot about the statistics of language. We were unable to produce comparable results with the standard GAN objective, though we do not claim that doing so is impossible.

Table 4: Samples from a WGAN character-level language model trained with our method on sentences from the Billion Word dataset, truncated to 32 characters. The model learns to directly output one-hot character embeddings from a latent vector without any discrete sampling step. We were unable to achieve comparable results with the standard GAN objective and a continuous generator.

| **WGAN with gradient penalty (1D CNN)** | |
| --- | --- |
| Busino game camperate spent odea | Solice Norkedin pring in since |
| In the bankaway of smarling the | ThiS record ( 31. ) UBS ) and Ch |
| SingersMay , who kill that imvic | It was not the annuas were plogr |
| Keray Pents of the same Reagun D | This will be us , the ect of DAN |
| Manging include a tudancs shat " | These leaded as most-worsd p2 a0 |
| His Zuith Dudget , the Denmbern | The time I paidOa South Cubry i |
| In during the Uitational questio | Dour Fraps higs it was these del |
| Divos from The ' noth ronkies of | This year out howneed allowed lo |
| She like Monday , of macunsuer S | Kaulna Seto consficutes to repor |

The difference in performance between WGAN and other GANs can be explained as follows. Consider the simplex $\Delta_n = \{p \in \mathbb{R}^n : p_i \geq 0, \sum_i p_i = 1\}$, and the set of vertices on the simplex (or one-hot vectors) $V_n = \{p \in \mathbb{R}^n : p_i \in \{0,1\}, \sum_i p_i = 1\} \subseteq \Delta_n$. If we have a vocabulary of size $n$ and we have a distribution $\mathbb{P}_r$ over sequences of size $T$, we have that $\mathbb{P}_r$ is a distribution on $V_n^T = V_n \times \cdots \times V_n$. Since $V_n^T$ is a subset of $\Delta_n^T$, we can also treat $\mathbb{P}_r$ as a distribution on $\Delta_n^T$ (by assigning zero probability mass to all points not in $V_n^T$).

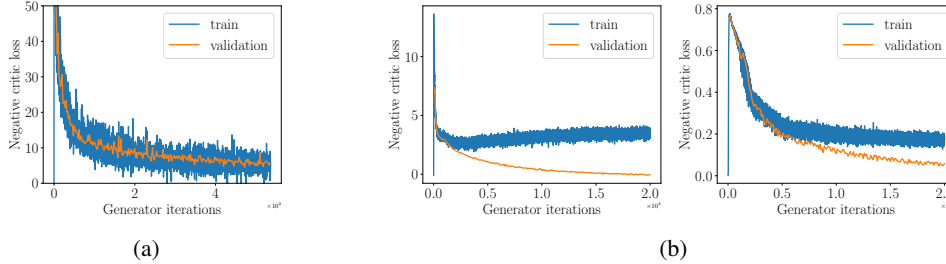

Figure 5: (a) The negative critic loss of our model on LSUN bedrooms converges toward a minimum as the network trains. (b) WGAN training and validation losses on a random 1000-digit subset of MNIST show overfitting when using either our method (left) or weight clipping (right). In particular, with our method, the critic overfits faster than the generator, causing the training loss to increase gradually over time even as the validation loss drops.

$\mathbb{P}_r$ is discrete (or supported on a finite number of elements, namely $V_n^T$) on $\Delta_n^T$, but $\mathbb{P}_g$ can easily be a continuous distribution over $\Delta_n^T$. The KL divergences between two such distributions are infinite, and so the JS divergence is saturated. In practice, this means a discriminator might quickly learn to reject all samples that don't lie on $V_n^T$ (sequences of one-hot vectors) and give meaningless gradients to the generator. However, it is easily seen that the conditions of Theorem 1 and Corollary 1 of [2] are satisfied even on this non-standard learning scenario with $\mathcal{X} = \Delta_n^T$. This means that $W(\mathbb{P}_r, \mathbb{P}_g)$ is still well defined, continuous everywhere and differentiable almost everywhere, and we can optimize it just like in any other continuous variable setting. The way this manifests is that in WGANs, the Lipschitz constraint forces the critic to provide a linear gradient from all $\Delta_n^T$ towards towards the real points in $V_n^T$.

Other attempts at language modeling with GANs [31, 14, 29, 5, 15, 10] typically use discrete models and gradient estimators [27, 12, 16]. Our approach is simpler to implement, though whether it scales beyond a toy language model is unclear.

### 5.6 Meaningful loss curves and detecting overfitting

An important benefit of weight-clipped WGANs is that their loss correlates with sample quality and converges toward a minimum. To show that our method preserves this property, we train a WGAN-GP on the LSUN bedrooms dataset [30] and plot the negative of the critic's loss in Figure 5a. We see that the loss converges as the generator minimizes $W(\mathbb{P}_r, \mathbb{P}_g)$.

GANs, like all models trained on limited data, will eventually overfit. To explore the loss curve's behavior when the network overfits, we train large unregularized WGANs on a random 1000-image subset of MNIST and plot the negative critic loss on both the training and validation sets in Figure 5b. In both WGAN and WGAN-GP, the two losses diverge, suggesting that the critic overfits and provides an inaccurate estimate of $W(\mathbb{P}_r, \mathbb{P}_g)$, at which point all bets are off regarding correlation with sample quality. However in WGAN-GP, the training loss gradually increases even while the validation loss drops.

[28] also measure overfitting in GANs by estimating the generator's log-likelihood. Compared to that work, our method detects overfitting in the critic (rather than the generator) and measures overfitting against the same loss that the network minimizes.

## 6 Conclusion

In this work, we demonstrated problems with weight clipping in WGAN and introduced an alternative in the form of a penalty term in the critic loss which does not exhibit the same problems. Using our method, we demonstrated strong modeling performance and stability across a variety of architectures. Now that we have a more stable algorithm for training GANs, we hope our work opens the path for stronger modeling performance on large-scale image datasets and language. Another interesting direction is adapting our penalty term to the standard GAN objective function, where it might stabilize training by encouraging the discriminator to learn smoother decision boundaries.

## Acknowledgements

We would like to thank Mohamed Ishmael Belghazi, Léon Bottou, Zihang Dai, Stefan Doerr, Ian Goodfellow, Kyle Kastner, Kundan Kumar, Luke Metz, Alec Radford, Sai Rajeshwar, Aditya Ramesh, Tom Sercu, Zain Shah and Jake Zhao for insightful comments.

## Footnotes

*Now at Google Brain

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
