[Supplementary Material]

# A Proof of Proposition 1

*Proof.* Since $\mathcal{X}$ is a compact space, by Theorem 5.10 of [24], part (iii), we know that there is an optimal $f^*$. By Theorem 5.10 of [24], part (ii) we know that if $\pi$ is an optimal coupling,

$$\mathbb{P}_{(x,y)\sim\pi}\left[f^*(y) - f^*(x) = \|y - x\|\right] = 1$$

Let $(x, y)$ be such that $f^*(y) - f^*(x) = \|y - x\|$. We can safely assume that $x \neq y$ as well, since this happens under $\pi$ with probability 1. Let $\psi(t) = f^*(x_t) - f^*(x)$. We claim that $\psi(t) = \|x_t - x\| = t\|y - x\|$.

Let $t, t' \in [0, 1]$, then

$$\begin{aligned}
|\psi(t) - \psi(t')| &= \|f^*(x_t) - f^*(x_{t'})\| \\
&\leq \|x_t - x_{t'}\| \\
&= |t - t'|\|x - y\|
\end{aligned}$$

Therefore, $\psi$ is $\|x - y\|$-Lipschitz. This in turn implies

$$\begin{aligned}
\psi(1) - \psi(0) &= \psi(1) - \psi(t) + \psi(t) - \psi(0) \\
&\leq (1 - t)\|x - y\| + \psi(t) - \psi(0) \\
&\leq (1 - t)\|x - y\| + t\|x - y\| \\
&= \|x - y\|
\end{aligned}$$

However, $|\psi(1) - \psi(0)| = |f^*(y) - f^*(x)| = \|y - x\|$ so the inequalities have to actually be equalities. In particular, $\psi(t) - \psi(0) = t\|x - y\|$, and $\psi(0) = f^*(x) - f^*(x) = 0$. Therefore, $\psi(t) = t\|x - y\|$ and we finish our claim.

Let

$$\begin{aligned}
v &= \frac{y - x_t}{\|y - x_t\|} \\
&= \frac{y - ((1 - t)x - ty)}{\|y - ((1 - t)x - ty)\|} \\
&= \frac{(1 - t)(y - x)}{\|(1 - t)\|y - x\|} \\
&= \frac{y - x}{\|y - x\|}
\end{aligned}$$

Now we know that $f^*(x_t) - f^*(x) = \psi(t) = t\|x - y\|$, so $f^*(x_t) = f^*(x) + t\|x - y\|$. Then, we have the partial derivative

$$\begin{aligned}
\frac{\partial}{\partial v} f^*(x_t) &= \lim_{h\to 0} \frac{f^*(x_t + hv) - f^*(x_t)}{h} \\
&= \lim_{h\to 0} \frac{f^*\left(x + t(y - x) + \frac{h}{\|y-x\|}(y - x)\right) - f^*(x_t)}{h} \\
&= \lim_{h\to 0} \frac{f^*\left(x_{t + \frac{h}{\|y-x\|}}\right) - f^*(x_t)}{h} \\
&= \lim_{h\to 0} \frac{f^*(x) + \left(t + \frac{h}{\|y-x\|}\right)\|x - y\| - (f^*(x) + t\|x - y\|)}{h} \\
&= \lim_{h\to 0} \frac{h}{h} \\
&= 1
\end{aligned}$$

If $f^*$ is differentiable at $x_t$, we know that $\|\nabla f^*(x_t)\| \leq 1$ since it is a 1-Lipschitz function. Therefore, by simple Pythagoras and using that $v$ is a unit vector

$$
\begin{aligned}
1 &\leq \|\nabla f^*(x)\|^2 \\
&= \langle v, \nabla f^*(x_t) \rangle^2 + \|\nabla f^*(x_t) - \langle v, \nabla f^*(x_t) \rangle v\|^2 \\
&= |\frac{\partial}{\partial v} f^*(x_t)|^2 + \|\nabla f^*(x_t) - v \frac{\partial}{\partial v} f^*(x_t)\|^2 \\
&= 1 + \|\nabla f^*(x_t) - v\|^2 \\
&\leq 1
\end{aligned}
$$

The fact that both extremes of the inequality coincide means that it was all an equality and $1 = 1 + \|\nabla f^*(x_t) - v\|^2$ so $\|\nabla f^*(x_t) - v\| = 0$ and therefore $\nabla f^*(x_t) = v$. This shows that $\nabla f^*(x_t) = \frac{y - x_t}{\|y - x_t\|}$.

To conclude, we showed that if $(x, y)$ have the property that $f^*(y) - f^*(x) = \|y - x\|$, then $\nabla f^*(x_t) = \frac{y - x_t}{\|y - x_t\|}$. Since this happens with probability 1 under $\pi$, we know that

$$
\mathbb{P}_{(x,y) \sim \pi} \left[ \nabla f^*(x_t) = \frac{y - x_t}{\|y - x_t\|} \right] = 1
$$

and we finished the proof.

$\square$

## B Experimental details and results for training random architectures within a set

Table 5: Outcomes of training 200 random architectures, for different success thresholds. For comparison, our standard DCGAN achieved a score of 7.24.

| Min. score | Only GAN | Only WGAN-GP | Both succeeded | Both failed |
|---|---|---|---|---|
| 1.0 | 0 | 8 | 192 | 0 |
| 1.5 | 0 | 50 | 150 | 0 |
| 2.0 | 0 | 60 | 140 | 0 |
| 2.5 | 0 | 74 | 125 | 1 |
| 3.0 | 1 | 88 | 110 | 1 |
| 3.5 | 0 | 111 | 86 | 3 |
| 4.0 | 1 | 126 | 67 | 6 |
| 4.5 | 0 | 136 | 55 | 9 |
| 5.0 | 0 | 147 | 42 | 11 |
| 5.5 | 0 | 148 | 32 | 20 |
| 6.0 | 0 | 145 | 21 | 34 |
| 6.5 | 1 | 131 | 11 | 57 |
| 7.0 | 1 | 104 | 5 | 90 |
| 7.5 | 2 | 67 | 3 | 128 |
| 8.0 | 1 | 34 | 0 | 165 |
| 8.5 | 0 | 6 | 0 | 194 |
| 9.0 | 0 | 0 | 0 | 200 |

All models were trained on $32 \times 32$ ImageNet for 100K iterations using Adam with hyperparameters as recommended in [21] ($\alpha = 0.0002, \beta_1 = 0.5, \beta_2 = 0.999$) for the standard GAN objective and our recommended settings ($\alpha = 0.0001, \beta_1 = 0, \beta_2 = 0.9$) for WGAN-GP.

## C Experiments with one-sided penalty

We considered a one-sided penalty of the form $\lambda \mathbb{E}_{\hat{x} \sim \mathbb{P}_{\hat{x}}} \left[ \max(0, \|\nabla_{\hat{x}} D(\hat{x})\|_2 - 1)^2 \right]$ which would penalize gradients larger than 1 but not gradients smaller than 1, but we observe that the two-sided

version seems to perform slightly better. We sample 174 architectures from the set specified in Table 1 and train each architecture with the one-sided and two-sided penalty terms. The two-sided penalty achieved a higher Inception score in 100 of the trials, compared to 77 for the one-sided penalty. We note that this result is not statistically significant at $p < 0.05$ and further is with respect to only one (somewhat arbitrary) metric and distribution of architectures, and it is entirely possible (likely, in fact) that there are settings where the one-sided penalty performs better, but we leave a thorough comparison for future work. Other training details are the same as in Appendix B.

## D   Nonsmooth activation functions

The gradient of our objective with respect to the discriminator's parameters contains terms which involve second derivatives of the network's activation functions. In the case of networks with ReLU or other piecewise linear activation functions, this means the gradient is undefined at some points (albeit a measure zero set), and the gradient penalty objective might not be continuous with respect to the parameters, causing optimization to fail. Empirically, this seems not to be a problem for some common activation functions: in our random architecture and LSUN architecture experiments we find that we are able to train networks with piecewise linear activation functions as well as smooth activation functions. We do note that we were unable to train networks with ELU activations, whose derivative is continuous but not smooth. Replacing ELU with a very similar nonlinearity which is smooth ($\frac{\text{softplus}(2x+2)}{2} - 1$) fixed the issue.

## E   Hyperparameters used for LSUN robustness experiments

- WGAN with gradient penalty: Adam ($\alpha = .0001, \beta_1 = .5, \beta_2 = .9$)

- WGAN with weight clipping: RMSProp ($\alpha = .00005$)

- DCGAN: Adam ($\alpha = .0002, \beta_1 = .5$)

- LSGAN: RMSProp ($\alpha = .0001$) [chosen by search over $\alpha = .001, .0002, .0001$]

## F   CIFAR-10 ResNet architecture

The generator and critic are residual networks; we use pre-activation residual blocks with two $3 \times 3$ convolutional layers each and ReLU nonlinearity. Some residual blocks perform downsampling (in the critic) using mean pooling after the second convolution, or nearest-neighbor upsampling (in the generator) before the second convolution. We use batch normalization in the generator but not the critic. We optimize using Adam with learning rate $2 \times 10^{-4}$, decayed linearly to 0 over 100K generator iterations, and batch size 64.

For further architectural details, please refer to our open-source implementation.

| Generator $G(z)$ | | | |
|---|---|---|---|
| | Kernel size | Resample | Output shape |
| $z$ | - | - | 128 |
| Linear | - | - | $128 \times 4 \times 4$ |
| Residual block | [ 3×3 ] × 2 | Up | $128 \times 8 \times 8$ |
| Residual block | [ 3×3 ] × 2 | Up | $128 \times 16 \times 16$ |
| Residual block | [ 3×3 ] × 2 | Up | $128 \times 32 \times 32$ |
| Conv, $tanh$ | 3×3 | - | $3 \times 32 \times 32$ |

| Critic $D(x)$ | | | |
|---|---|---|---|
| | Kernel size | Resample | Output shape |
| Residual block | [ 3×3 ] × 2 | Down | $128 \times 16 \times 16$ |
| Residual block | [ 3×3 ] × 2 | Down | $128 \times 8 \times 8$ |
| Residual block | [ 3×3 ] × 2 | - | $128 \times 8 \times 8$ |
| Residual block | [ 3×3 ] × 2 | - | $128 \times 8 \times 8$ |
| ReLU, mean pool | - | - | 128 |
| Linear | - | - | 1 |

# G  CIFAR-10 ResNet samples

Figure 6: *(left)* CIFAR-10 samples generated by our unsupervised model. *(right)* Conditional CIFAR-10 samples, from adding AC-GAN conditioning to our unconditional model. Samples from the same class are displayed in the same column.

# H    More LSUN samples

Method: DCGAN
$G$: DCGAN, $D$: DCGAN

Method: DCGAN
$G$: No BN and const. filter count

Method: DCGAN
$G$: 4-layer 512-dim ReLU MLP

Method: DCGAN
No normalization in either $G$ or $D$

Method: DCGAN
Gated multiplicative nonlinearities

Method: DCGAN
$tanh$ nonlinearities

Method: DCGAN
101-layer ResNet $G$ and $D$

Method: LSGAN
$G$: DCGAN, $D$: DCGAN

Method: LSGAN
$G$: No BN and const. filter count

Method: LSGAN
$G$: 4-layer 512-dim ReLU MLP

Method: LSGAN
No normalization in either $G$ or $D$

Method: LSGAN
Gated multiplicative nonlinearities

Method: LSGAN
$tanh$ nonlinearities

Method: LSGAN
101-layer ResNet $G$ and $D$

Method: WGAN with clipping
$G$: DCGAN, $D$: DCGAN

Method: WGAN with clipping
$G$: No BN and const. filter count

Method: WGAN with clipping
$G$: 4-layer 512-dim ReLU MLP

Method: WGAN with clipping
No normalization in either $G$ or $D$

Method: WGAN with clipping
Gated multiplicative nonlinearities

Method: WGAN with clipping
$tanh$ nonlinearities

Method: WGAN with clipping
101-layer ResNet $G$ and $D$

Method: WGAN-GP (ours)
$G$: DCGAN, $D$: DCGAN

Method: WGAN-GP (ours)
$G$: No BN and const. filter count

Method: WGAN-GP (ours)
$G$: 4-layer 512-dim ReLU MLP

Method: WGAN-GP (ours)
No normalization in either $G$ or $D$

Method: WGAN-GP (ours)
Gated multiplicative nonlinearities

Method: WGAN-GP (ours)
$tanh$ nonlinearities

Method: WGAN-GP (ours)
101-layer ResNet $G$ and $D$