[Reviews · NeurIPS 2017]

Reviewer 1



+ good results + pretty pictures - exposition could be improved - claims are not backed up - minor technical contribution - insufficient evaluation Let me first say that the visual results of this paper are great. The proposed algorithm produces pretty pictures of bedrooms and tiny cifar images. Unfortunately the rest of the paper needs to be significantly improved for acceptance at NIPS. The exposition and structure of the paper is below NIPS standards. For example L3: "..., but can still generate low-quality samples or fail to converge in some settings." I'm assuming the authors mean "can only..."? l6: "pathological behavior". Pathological seems the wrong word here... Sections 3 and 4 should be flipped. This way the exposition doesn't need to refer to a method that was not introduced yet. Secondly, the paper is full of claims that are not backed up. The reviewer recommends the authors to either remove these claims or experimentally or theoretically back them up. Here are a few: L101: Proof that only simple functions have gradient norm almost everywhere, note that almost everywhere is important here. There can be exponentially many parts of the landscape that are disconnected by gradient norm!=1 parts. If a proof is not possible, please remove this claim. L 144: Two sided penalty works better. Please show this in experiments. The actual technical contribution seems like a small modification of WGAN. Box constraint is relaxed to a l2 norm on gradients. Finally the evaluation is insufficient. The paper shows some pretty generation results for various hyper-parameter setting, but doesn't show any concrete numeric evaluation, apart from the easy to cheat and fairly meaningless inception score. The authors should perform a through user study to validate their approach, especially in light of a slim technical contribution. Minor: How is (3) efficiently evaluated? Did the authors have to modify specific deep learning packages to compute the gradient of the gradient? How is this second derivative defined for ReLUs? Post rebuttal: The authors answered some of my concerns in the rebuttal. I'm glad the authors promised to improve the writing and remove the misleading claims. However the rebuttal didn't fully convince me, see below. I'm still a bit hesitant to accept the argument that prettier pictures are better, the authors rebuttal didn't change much in that regard. I'm sure some image filter, or denoising CNN could equally improve the image quality. I would strongly suggest the authors to find a way to quantitatively evaluate their method (other than the inception score). I'm also fairly certain that the argument about the second derivative of ReLU's is wrong. The second derivative of a relu is not defined, as far as I know. The authors mention it's all zero, but that would make it a linear function (and a relu is not linear)... The reviewers had an extensive discussion on this and we agree that for ReLU networks the objective is not continuous, hence the derivative of the objective does not point down hill at all times. There are a few ways the authors should address this: 1. Try twice differentiable non-linearities (elu, tanh or log(1+exp(x)). 2. Show that treating the ReLU as a linear function (second derivative of zero) does give a good enough gradient estimate, for example through finite differences, or by verifying how often the "gradient" points down hill.

Reviewer 2



The authors suggest a simple alternative to the weight clipping of Wasserstein GAN. Although presentation could be greatly improved, it is a decent contribution with practical use. Please implement the following changes for reviewers to better evaluate the soundness of the claims. The result given in Section 2.3 is not sufficiently explained. Please elaborate on how this result helps explain the problem with weight clipping. Also the claim in Line 85 is not implied by the paragraph above. It may help to formalize the paragraph statement in a Proposition and write Linen 85 as a corollary. Line 99 then can refer to this corollary. The same claim is also used in Line 130. Hence it is very important to clarify this claim. It would be useful to see the effect of batch normalization on the Swiss roll data in Figure 1b. Is the result as good as gradient penalty? Please consider adding this to the figure. AFTER AUTHOR FEEDBACK AND REBUTTAL: The authors have made the mistake of claiming (in their feedback) that the second derivative is nonzero because they do not take the second derivative of ReLU wrt a variable but first wrt input variables and then wrt weights. Unfortunately this is not exactly correct. Thanks to the careful observation of R3, in a simple 1 layer ReLU network, the second derivative of ReLU wrt its input comes up as an additive term even then. This term will be zero in almost all local neighborhoods, but more importantly, it will not be defined simply because second derivative of ReLU wrt its input does not exist since the first derivative is not continuous. It is not clear what will happen with more layers and I think this analysis would be a nice future direction. I suggest acceptance of the paper. But I ask reviewers to implement the following change: Present your results using tanh as the nonlinearity, which lets you define first and second order derivatives without any mathematical mistake. Later, explain the audience that the same result holds for ReLU networks in practice even though second derivative is not defined and try to justify why (for example local derivatives should work unless you land on finitely many "bad" points in practice). Adding the results showing nonzero ReLU gradient which the authors referred to in their feedback could help.